# Treating Racial Trauma: The Methodology of a Randomized Controlled Trial of the Healing Racial Trauma Protocol

**DOI:** 10.3390/bs15070856

**Published:** 2025-06-25

**Authors:** Muna Osman, Sophia Gran-Ruaz, Lucia Rios Maia da Silva, Monnica T. Williams

**Affiliations:** School of Psychology, University of Ottawa, Ottawa, ON K1N 6N5, Canada; mosma039@uottawa.ca (M.O.); sgran089@uottawa.ca (S.G.-R.); lrios085@uottawa.ca (L.R.M.d.S.)

**Keywords:** racial trauma, randomized controlled trial, clinical trial, racism, racialized

## Abstract

Cumulative experiences of racism lead to stress and trauma. Racial trauma is associated with compromised functioning across psychological, social, and physical health domains. This is further complicated by any comorbidity with other mental health conditions. Many clinicians are not trained in identifying, diagnosing, and treating racial trauma. Given the pervasive nature of racism, limited clinician knowledge and experiences, as well as the impact of this condition, there is an urgent need for novel, culturally safe, and effective treatment. The newly developed Healing Racial Trauma Protocol (HRTP) shows significant promise. We explore the methodological considerations for a randomized controlled trial comparing the efficacy of the HRTP for racialized individuals suffering from racial trauma and to a control condition of treatment as usual, in reducing the severity of racial trauma and depression symptoms, as well as improved functioning. Ethical, pragmatic, and methodological considerations in trial design, research population, and treatment protocol are explored.

## 1. Introduction

People of colour experience a range of health disparities rooted in the consequences of racism ([10]; [53]). These health disparities are further exacerbated by challenges in access to timely, quality, and culturally safe healthcare services, including prevention and treatment. *Racial trauma* refers to the traumatic effects of individual and systemic racism over time, which can often match the severity and symptomatology of posttraumatic stress disorder (PTSD; [9]; [60]). The early detection, diagnosis, and treatment of racial trauma is critical to mitigate the deleterious effects of racism on quality of life, chronic health risks, and social relationships. Evidence for the efficacy of conventional PTSD treatment to address trauma in people of colour is limited due to the lack of ethnoracial representation in research, poor reporting standards for race-based data, and the limited ethnoracial diversity of clinicians ([19]). Therefore, evidence of novel, culturally safe, and effective treatment approaches is essential. Specifically, a targeted clinical trial can provide empirical evidence for the efficacy of a culturally safe racial trauma-specific treatment compared to a conventional treatment approach. This paper outlines the methodological considerations of the trial design, research population, and treatment protocols for a randomized controlled trial for the treatment of racial stress and trauma for Canadians of colour.

### 1.1. Mental Health Disparities Among Racialized Communities

There has been a steady increase in Canada’s racialized population due to increasing population growth and immigration. The population of racialized Canadians increased by 130% from 2001 to 2021 compared to a 1% increase among White Canadians ([42]). Two thirds of this growth are attributed to the arrival of new immigrants, while one third is attributed to a Canadian-born population ([42]). In 2021, 5% of the population in Canada was Indigenous, and one in four Canadians (26.5%) were racialized, reaching a population of 9 million. From the racialized groups, South Asian people represented 26%, Chinese people 18%, and Black people 16% ([42]). Racialized communities reflect diverse cultures, social identities, and experiences. Race and ethnicity often intersect with other social identities based on gender, sexuality, disability, language, socioeconomic circumstances, and education. For many racialized groups, the process of racism leads to their differential treatment due to people’s perception of their shared appearance and presumed ancestry ([24]). Racism impacts mental health disparities through access to opportunities (e.g., employment, housing, education), exposure to risk factors (e.g., environmental risk, violence), patterns in cognitive and emotional processes associated with psychopathology, and behaviours related to coping and self-regulation ([25]; [30]).

Many racialized Canadians experience health disparities in both physical and mental health conditions rooted in racism and systemic oppression, which is further exacerbated by limited access to timely and quality healthcare ([19]; [25]). Racialized communities face increased risk and diagnosis of specific mental health conditions, while also being underdiagnosed for other conditions ([11]; [33]; [18]). These experiences are further exacerbated by additional barriers, including financial constraints, fear of discrimination from service providers, and a lack of representation among mental health professionals ([19]). Further, inequities in conditions and access are more pronounced for people with intersecting identities ([57]).

Lastly, the lack of race-based data in Canada makes it harder to address these disparities, leaving racialized individuals without adequate support. Only recently has Canada prioritized the collection and analysis of race-based data to examine disparities in life conditions, health inequities, and access to care ([39]; [58]). However, many health and social services institutions do not routinely collect data on race and ethnicity to examine disparities in health conditions and access. This is further hindered by the lack of data on intersecting identities, such as immigration status, language minorities, region, gender, and disability, among others. The collection and analysis of disaggregated sociodemographic data is essential to assess and address disparities at a granular level and make more informed decisions.

#### Racial Stress and Trauma

Among the mental health inequities faced by Indigenous peoples and racialized communities, racial trauma is a condition associated with negative impacts across psychological, social, and physical health domains ([9]; [60]). Despite Canada’s commitments to multiculturalism and inclusion, systemic racism and discrimination create significant stress. Racial trauma results from direct or repeated experiences of racism, including microaggressions, exclusion, and institutional discrimination. This condition can also co-occur with other mental health challenges, including anxiety, suicidal ideation, depression, despair, and poor physical health ([4]; [60]), which can go undiagnosed and often unrecognized by mainstream mental health providers. Although many therapists report serving clients with symptoms of racial trauma, few have sufficient training in the assessment and treatment of this condition ([23]).

Several theoretical frameworks have identified the pathways linking racism to poor mental health in general. However, few models have examined the link between racism and racial trauma. Williams and colleagues proposed a framework to understand the mechanisms that link the cumulative effects of racism with racial trauma, which often manifests through symptoms of PTSD ([54], [55], [52]). This model identified risk factors associated with historical and cultural trauma and race-based oppression, as vulnerabilities that interact with cumulative life experiences related to covert and overt racism. For a racialized person, racially charged traumatic events can result in intense emotional responses of shock, fear, and anger. When these experiences are invalidated, across time and people, PTSD-like symptoms can develop including intrusive thoughts, avoidance, negative changes in mood and cognitions, as well as changes in physical and emotional arousal. With time, these experiences are exacerbated by limited access and barriers to quality and timely care due to institutional racism. That is, a care provider might have limited knowledge of or a reluctance to address issues of race and racism in the therapeutic setting.

Based on this framework and the existing literature, researchers have identified key components essential to treating racial trauma ([7]; [12]; [14]; [35]; [2]), yet, there still is no adapted CBT treatment specific for racial trauma with evidence of efficacy through a randomized clinical trial.

### 1.2. A Promising New Intervention: The Healing Racial Trauma Protocol

Although racial stress and trauma are common, there are few evidence-based manualized treatments available for people of colour suffering from racial trauma ([56]). *The Healing Racial Trauma Protocol* (HRTP) is an innovative treatment protocol that provides a detailed approach on how to help people suffering from racial stress and trauma through guided content over 12 sessions ([49]). This therapy comprises a total of 12 sessions (each session is 1 h), delivered weekly over a period of 3 months. Grounded in cognitive behavioural therapy (CBT), as well as the third-wave behavioural approaches of Functional Analytic Psychotherapy (FAP) and Acceptance and Commitment Therapy (ACT), this protocol has three sections. The first section is Stabilization and Support, which consists of the first three sessions. The second is Healing the Wounds of Racism, which includes the next 4–5 sessions, and the third section is Empowerment, which includes the last 4 sessions. The structure of these sections is similar to Herman’s three stages of psychological trauma recovery (2015), which include establishing safety, retelling the story of the traumatic event, reconnecting with others, and empowerment.

The HRTP is culturally safe primarily because it recognizes that the most significant source of therapeutic safety is not the protocol itself, but rather the clinician who delivers it. Although the techniques used in the HRTP have been vetted through research studies and reflect widely accepted therapeutic approaches (e.g., CBT, FAP, and ACT; [24]; [52]), these methods alone cannot ensure cultural safety. Instead, the clinician’s capacity to create a safe and validating environment—through ongoing education, cultural competency training, and personal anti-bias work—is what ultimately fosters a secure therapeutic space ([51]). This emphasis on the clinician’s role acknowledges the importance of addressing power dynamics, historical injustices, and the lived experiences of racialized clients in the therapeutic process.

In practice, the HRTP demands that clinicians be mindful of how systemic and individual biases can emerge, even within well-intentioned therapeutic contexts. By requiring that clinicians reflect on their own biases, develop cultural humility, and engage in continuous learning, this ensures an appropriate base level of safety for participants. It exemplifies an approach in which therapeutic exercises are informed by evidence-based principles and also tailored to honour and validate each client’s cultural identity and lived reality. In this way, the HRTP situates itself within a framework of cultural safety that moves beyond specific CBT techniques, centering relational factors that are critical to the healing process.

### 1.3. Need for More Inclusive Clinical Research

Clinical trials are widely regarded as the gold standard for establishing evidence-based clinical treatment options. They play a critical role in evaluating the efficacy of new therapies, like the HRTP, as well as providing specific methodological frameworks to guide practice. Novel treatment options, in particular, depend on rigorous clinical trial data to demonstrate their safety and effectiveness. However, to date, racialized populations have been disproportionately excluded from such trials, skewing the evidence base and limiting the generalizability of findings (e.g., [32]; [45]). This is a pressing concern, as many racialized groups—including Black and Indigenous populations—experience disproportionately high rates of trauma ([21]). Yet they remain underrepresented in PTSD and other mental health research ([21]; [34]). Consequently, entire approaches intended to alleviate suffering and address mental health concerns have historically omitted large segments of racialized communities ([28]).

Several factors contribute to the exclusion of racialized groups from clinical trials, all of which are rooted in systemic and structural racism. First, many racialized people harbour deep mistrust of the medical and research establishment, a sentiment driven by well-documented historical abuses and continued experiences of medical racism (e.g., [27]; [43]). Second, researchers frequently operate under biassed assumptions that racialized individuals will not be reliable participants, leading to higher exclusion rates (e.g., [31]). Finally, the lack of culturally informed methodologies hinders effective recruitment and retention of diverse groups, ultimately reinforcing existing disparities in mental health research (e.g., [46]; [48]; [50]). Addressing these systemic barriers is essential for developing inclusive, equitable, and culturally informed approaches to treating racial trauma.

### 1.4. The Goals of the Current Study

The primary objective of this paper is to describe a randomized controlled trial that will compare the efficacy of the HRTP for people of colour suffering from racial trauma compared to a control condition of treatment-as-usual (TAU) in reducing the severity of racial trauma and related symptoms, as well as improving functioning. This paper will provide preliminary feasibility evidence in terms of implementation, practicality, and acceptability for clients and therapists. Additionally, we will outline methodological approaches to trial design, the research population, and treatment protocols. That is, methodological approaches which could serve as inclusive and viable models for engaging in clinical research related to racialized populations. Specifically, the proposed trial will examine: (a) whether the administration of the HRTP will result in a clinically significant reduction in racial trauma symptoms in participants compared to the TAU control group, and (b) whether any clinically significant reduction in racial trauma symptoms is maintained long-term (i.e., 4 months after the intervention).

It is hypothesized that the completion of the 12-session HRTP will result in a clinically significant improvement in participants’ racial trauma symptoms and severity, as well as improved functioning. It is also hypothesized that these changes will be greater than changes in the control conditions and will be maintained 4 months following the completion of the intervention.

## 2. Methodology

### 2.1. Trial Design and Setting

A parallel-group, single-blinded, two-arm randomized controlled superiority trial compares the effectiveness of HRTP to TAU with a 1:1 allocation ratio. Quantitative and qualitative data will be used to examine the effect of HRTP on racial trauma, depression (primary outcomes), and functioning (secondary outcomes). The trial will take place in an urban city in the province of Ontario, Canada.

Conducting the study in Ottawa contributes to the generalizability of the findings due to the city’s demographic and cultural diversity. As Canada’s capital, Ottawa attracts individuals from across the country and around the globe, providing a considerable mix of racial, ethnic, and linguistic communities. Based on the 2021 Population Census, Black people represented the largest racialized group at 8.5%, while Arabs and South Asian people each represented 5.8% of the total population of the city ([40]). As of 2021, there are also some 46,500 Indigenous people living in the Ottawa area, making up 3.2% of the population ([40]). This diversity means that the study is more likely to capture a range of lived experiences, cultural contexts, immigration histories, and socioeconomic backgrounds, thereby strengthening the external validity of the findings. A broad and varied participant pool will help ensure results are applicable to other urban centres and potentially even to more rural or international settings that share similar multicultural characteristics.

This study will follow the ethical principles of the Canadian Code of Ethics for Psychologists ([8]) and the Ethical Conduct for Research Involving Humans ([44]). Informed consent will be obtained from all participants. The pilot study has been approved by the Research Ethics Board (H-02-22-7613) of the University of Ottawa, ensuring the protection of participant privacy and confidentiality. Participants may withdraw from the study at any time without consequence.

### 2.2. Sample Size, Randomization, and Allocation

A sample of 96 participants (48 per group) is needed for an 80% probability to identify a difference between the treatment and control group. The power analysis for the sample size estimation was based on the means, standard deviations, and clinical cut-off for the primary outcomes (RTS, TSDS; [55], [57]). Drawing on previous research, we hypothesized a standard mean difference of between 0.36 and 0.52, between the treatment and control group, which translates to a 10–20% decrease in the primary outcomes (i.e., RTS, TSDS) for individuals at the endpoint of treatment. Given these effect estimates, with an alpha of 0.05, an 80% power, and a 20% attrition, we estimated a sample size of approximately 96 (48 per group) will be needed. This randomized controlled trial will use an ANOVA and t-test (alpha error level = 5%, two-sided) for independent samples.

This sample size will require tailored recruitment efforts, community partnerships, and culturally safe retention strategies throughout the trial. Given differential attrition rates for racialized communities, the research team will spend additional time at intake discussing barriers to participation, issues with attendance, and address any participant concerns. Additionally, the research team will offer flexible session meeting times (e.g., evenings and weekends), and telephone check-ins as needed. The research team shall also compensate participants for both their completed sessions and batteries of measures.

In terms of blinding, the trial research team will be unaware of the group allocation. This includes the principal investigator and project coordinator who are overseeing the trial implementation, research assistants who will support recruitment and data collection, and the statistician conducting the analysis. See Figure 1 for the participant flow diagram of the trial.

### 2.3. Participant Eligibility, Recruitment, and Intake

*Participants* will be eligible for inclusion as *clients* if they have clinically relevant racism-related traumatic stress as determined by the UnRESTS (see the “Intake Measures for Inclusion” for more details), are at least 18 years of age, identify as Black, Indigenous, or a person of colour in Canada (BIPOC), have written and oral proficiency in English or French, and have access to the internet for online sessions. The only exclusion criterion includes having severe current substance or alcohol use as identified via the Mini International Neuropsychiatric Interview. However, participants who partake in another psychotherapy treatment or intervention will be advised to notify their external therapists about their participation in the study.

Participants will be recruited through the research team’s vast existing network of research partners and community mental health agencies. For example, the authors will partner with an academic research centre dedicated to the study of the biopsychosocial determinants of health in racialized, and specifically, Black communities in Canada. The centre’s existing participant pool and broad health sector network, including hospitals, community health centres, and community and religious organizations, will be leveraged. Mental health clinics within the study’s catchment will also be made aware of the study and invited to refer clients, when appropriate. Moreover, as is advised, to help build relationships and awareness of racial trauma, the HRTP, and the study, select offers will be sent to relevant associations (e.g., Ottawa Academy of Psychology) to conduct free/reduced-cost workshops/presentations for local mental health professionals ([48]). Interested individuals from the general public may also refer themselves to the study. These recruitment strategies can introduce selection bias; however, this bias is a common limitation in similar psychotherapy treatment studies (e.g., [20]).

Moreover, the research team will use targeted emails, social media posts, local bus ads, and word of mouth to advertise the study, instructing interested individuals to reach out to a study-specific email address. Once interest is confirmed, an email response will be sent with detailed information about the study and a consent form for consideration. The said study-specific email address will also present a direct contact with whom prospective participants may communicate should they have any questions or concerns about the study. Those interested in moving forward will be invited to partake in a 3 h virtual intake session (See Table 1 for the participants’ assessment schedule). During this intake, a trained research assistant will (i) describe important elements of the study, including the participant’s role and responsibilities; (ii) collect the necessary information from the participant to start a “file” (e.g., date of birth, gender, occupation, ethnoracial identity, contact information, emergency contact); (iii) and administer two semi-structured diagnostic interviews (i.e., the Mini International Neuropsychiatric Interview as well as the University of Connecticut Racial/Ethnic Stress & Trauma Survey). A 15 min break will be provided at the halfway point of the intake session. Using this intake information, the research team will continually and iteratively assess the demographic characteristics of the enrolled participants, gender and otherwise, to ensure sample heterogeneity and the generalizability of the findings (e.g., age, ethnicity, immigrant status, and socioeconomic status). Participant personal information and identifiers will be stored in a data management system to ensure participant privacy protection and data security. Data used for analysis will only include unique identification numbers assigned to each participant at intake. The data will be published in aggregate form and no participant identifier will be included.

Following intake, the participants will be randomly assigned to one of two groups: the HRTP (intervention) group or the control group (TAU). The randomization will be conducted using a computer-generated randomization sequence with stratification by age, gender, and the baseline level of racial stress (via the Racial Trauma Scale; [53]).

This trial will use a stratified randomization process to incorporate study covariates (i.e., age, gender, and racial stress levels) each with 2 categories for a total of 6 strata. While transgender and gender non-conforming people account for 0.3% of Canada’s population ([41]), the trial will oversample this population because of low attrition rates and the reality of gender minority populations being traditionally excluded in clinical research ([1]; [29]). We will evaluate and report any gender differences in response to treatment and outcomes through subgroup analysis to consider the intersectionality of minoritized individuals (e.g., [1]).

### 2.4. Participant Retention Strategies

The research team will use a range of equitable and inclusive strategies to ensure participant retention and mitigate attrition throughout the study. Once participants have completed the intake process and are fully onboarded, they will be assigned to a treatment group and a therapist. Through the sessions, the research team, including the therapists, will work with participants to sustain their participation and address barriers to their participation as they emerge. Participants will receive reminders based on their contact preferences, as well as notifications on our platform, for information about their sessions, questionnaires to complete, and any updates from the therapist and research team. Weekly sessions will be scheduled collaboratively with participants with flexibility incorporated into the available hours, and with the scheduling and rescheduling of sessions as needed. This flexibility will ensure therapists can identify and address any relevant barriers to a participant’s involvement. If a participant misses two sessions without further follow-ups with the research team, they will be invited to reconsider a better time to join the project. Further, participants will be paid for each session and packet of measures completed. For instance, within a pilot study of the HRTP trial, participants received $5 for each session as well as $30 for each battery of measures completed (e.g., intake and week 4).

The approach and strategies of this study are specific to racialized communities recognizing the historical and contemporary barriers to research access and participation. Specifically, we recognize participants will need explicit information and engagement as many might not have prior experience of participating in a clinical treatment study. Additional effort and time will be taken to answer participant questions, explain the process of their participation, and flexibility will be built into the design to support their continued involvement. These methodological considerations are essential to working with racialized communities experiencing racial trauma and the design and implementation are deliberate to centre the concerns of these communities to ensure their equitable and inclusive participation.

As an additional equitable measure, upon completion of the TAU control arm and the 4-month follow-up, participants will be offered the opportunity to receive the HRTP treatment arm independent of their trial participation.

### 2.5. Therapist Eligibility, Training, and Supervision

Given the innovative nature of the HRTP, extensive onboarding and training will be required to ensure therapists are competent and confident in terms of delivering the protocol. Given that many therapists may not have the necessary skills and training to work extensively with racialized communities, a selection, training, and supervision process was designed to supplement knowledge gaps in order to ensure therapists are comfortable discussing race and racism, can provide psychoeducation on race-based stressors, and possess the knowledge to address the racialized aspects of traumatic experiences ([7]). *Therapists* are eligible for inclusion in this study if they are psychology doctoral students with at least one full year or more of advanced clinical training (i.e., practicum or internship level). Therapists will primarily be recruited through the authors’ host institution’s community mental health clinic. Note, at any given time within the year, approximately 50 doctoral students provide psychological services through the said clinic. Also of note, therapists within this study will be compensated for direct/participant-facing time accrued. That is, HRTP and TAU therapists alike will be paid for each completed session with a participant.

*Supervisors* are eligible for inclusion in this study if they (i) are licenced clinical psychologists and (ii) have previous experience supervising clinical trainees. Moreover, supervisors overseeing therapists within the TAU control arm of the study must be certified or experienced to work within the clinic delivering the TAU therapeutic sessions. Similarly, those supervisors supporting therapists within the HRTP intervention arm of the study must be experienced in working with real-world clients suffering from racial trauma. Whether supporting therapists within the HRTP or TAU arms of this study, supervisors will be paid for each hour spent in supervision.

*HRTP therapists* will partake in a multistep onboarding process before they are permitted to see clients. Moreover, depending on the therapists’ level of training and/or experience working with racialized clients, the onboarding process will be adapted to increase therapist comfort and efficacy.

As a first step of the onboarding process, interested parties will be required to respond to a standard set of questions (see [51] for the full complement of questions). Sample questions from this set include:Have you had at least a one-semester graduate course (or the equivalent) focused on multicultural counselling skills? How have you applied it in your professional work?Are you willing to address racial differences with clients early in therapy? Give an example of how you would approach this.How would you respond if your client said you were a racist? Given that this question may require more context, assume that the client says this because you are either White or not the same race as they are.

Responses to the aforementioned questions will be used to gauge therapist interest, strengths, and gaps in training as well as to evaluate clarity, understanding of racism, and how it occurs in different settings. The Project Coordinator, and at least one other HRTP supervisor, will also review each submission against a study-specific rubric and follow up as needed on unclear elements or perceived gaps in responses. For instance, therapists may be assigned additional readings or role-play activities when their initial answers are insufficient.

In a second step, the therapists will be given mandatory material for review. Therapists with minimal clinical experience overall or in working with racialized clients will be required to read and reflect on two academic articles. Specifically, one article describing the character traits and behaviours of an anti-racist clinician (i.e., [51]); and another describing the underlying mechanisms employed by the HRTP (i.e., [52]). All HRTP therapists, regardless of experience level, must also watch three workshop videos recorded by the principal investigator, presenting each of the three phases of treatment in depth, with interactive case examples. To ensure therapists have thoroughly engaged with the material given, the therapists will prepare a short written or oral reflection on each article/video assigned. In so doing, the therapists will be asked to consider what they liked about the reading/video (and why), what they did NOT like about the reading/video (and why), and what impact the reading/video has on theory and practice. Again, the Project Coordinator and at least one other HRTP supervisor shall review the reflections, using another study-specific rubric, and provide commentary and guidance as needed.

Should the above two phases be completed smoothly and absent any issue, the therapists will be ready to be matched with a client. If the Project Coordinator and at least one other HRTP supervisor believe the individual requires extra support, a mock first session will be organized between the therapist and a peer. The Project Coordinator will provide feedback on the audio recording of said role play.

The *TAU therapists* will also partake in minimal onboarding activities including reviewing the trial protocol and an orientation at the control group’s treatment clinic, at the authors’ host institution. Onboarding for both HRTP and TAU therapists occurs before the said individuals see their first client. Nevertheless, training for both the intervention and control arms will not stop there. Once actively seeing clients, study therapists will be required to partake in weekly group supervision meetings. These meetings will last up to 90 min and be led by licenced clinical psychologists with general clinical experience (if a TAU supervision) or the HRTP (if an intervention supervision). Several supervision groups will run simultaneously to ensure group numbers remain low. That is, no more than 15 active therapists will be assigned to a single group supervision at a time. Moreover, while admission to supervision groups will be staggered (i.e., client and therapist onboarding-dependant), once placed, therapists will typically remain within their assigned group until they no longer wish to be part of the study (for case continuity and to more easily monitor therapists’ clinical goal progress).

### 2.6. HRTP Therapy (Intervention)

As mentioned above, the HRTP is a novel approach for the treatment of racial trauma, with roots in CBT, FAP, and ACT ([49]). The said protocol consists of 12 sessions, each 60 min in length, spanning three stages: (1) Stop the Bleeding–Stabilization and Support; (2) Healing the Wounds of Racism; and (3) Empowerment. Within phase 1 or “Stabilization”, therapists support their clients in understanding the wide-reaching effects of racism as well as racism’s relationship to racial trauma and other related conditions. As such, these early sessions are rich in psychoeducation and the thoughtful validation of each client’s experiences of racism and their reaction to these events. The therapist also helps the client make use of functional coping strategies against racism, with attempts made to increase the client’s self-care practices and enlarge their social support network.

Within phase 2 or “Healing”, the therapist moves towards processing the client’s upsetting experiences of racism. To start, the therapist guides the client in exploring and dismantling any internalized racism, offering genuine affirmations to expedite the process. The topic of colorism is also explored in great depth, and steps are taken to strengthen the client’s ethnoracial identity via psychoeducation and cultural engagement. Following this, two sessions are dedicated to having therapists bear witness as clients recount and habituate to racism-related traumatic experiences via imaginal exposures and/or other creative means (e.g., written expression and photovoice). To facilitate processing, therapists validate and reflect back client experiences, identify any negative cognitions about the client’s self-view that are reinforced by the traumatic experience, and offer a possible counternarrative. Finally, the therapists introduce microinterventions, role-playing, and imagery rescripting as tools to combat racism in various scenarios.

In phase 3 or “Empowerment”, the therapist supports clients in positively transforming their values, actions, and relationships in the wake of their past traumatic experiences (i.e., achieving posttraumatic growth). The therapists will also help their clients explore what their experiences mean to them (i.e., meaning-making). The final sessions of the HRTP consider how participating in social action and activism can also lead to continued healing. A last session will be used to review key concepts, celebrate gains, and discuss a plan to maintain wellness in the face of continued racism post therapy.

### 2.7. Treatment as Usual (Control)

Given the lack of validated alternative treatments for the healing of racial trauma ([56]), a TAU control group will be used. As such, participants will be referred to treatment at the authors’ host institution’s community mental health clinic after the completion of the intake assessment. The said clinic employs a variety of evidence-based therapeutic approaches, including CBT, emotionally focused therapy, existential therapy, interpersonal therapy, and psychodynamic therapy. Participants within the TAU group will receive 12 sessions of one of the said therapies (supervisor dependent).

### 2.8. Study Measures

The study measures included in the trial are two intake measures, three primary outcome measures, and two secondary outcome measures. Before study admission, two intake measures will be administered to prospective participants. These measures will be used to assess if prospective participants meet the study’s inclusion criteria and to better understand the participants’ mental health profiles.

The *Mini International Neuropsychiatric Interview* (MINI 7.0.2; [38]) is a semi-structured diagnostic interview designed to briefly assess 17 of the most common mental disorders according to the *Diagnostic and Statistical Manual of Mental Disorders, Fifth Edition* (DSM-5) criteria. The modules assessed include major depressive episodes, suicidality, manic and hypomanic episodes, panic disorder, agoraphobia, social phobia, obsessive–compulsive disorder, post-traumatic stress disorder, alcohol use disorder, substance use disorder, psychotic disorders and mood disorders with psychotic features, anorexia nervosa, bulimia nervosa, binge eating disorder, and generalized anxiety disorder, among others. The MINI was initially created for cross-cultural application and has since been adapted for use in several languages and diverse populations worldwide (e.g., [13]; [17]).

The *University of Connecticut Racial/Ethnic Stress & Trauma Survey* (UnRESTS; [54]) is a semi-structured diagnostic interview for assessing racial trauma. It consists of several parts, including probes around an individual’s racial and ethnic identity development, experiences of direct overt racism, experiences of racism by loved ones, experiences of vicarious racism, and experiences of covert racism (or microaggressions). The interview concludes with a 29-item checklist which assesses whether an individual’s racial trauma satisfies a DSM-5 diagnosis for PTSD. To render a racial trauma diagnosis ([60]), an individual must satisfy the criteria for a minimum of three of the four PTSD symptom clusters as per the checklist (i.e., re-experiencing, avoidance, negative changes in cognition and mood, and physiological arousal and reactivity). The said symptoms must also be clinically significant and must have been present for a month or more.

### 2.9. Primary Outcomes

Three primary outcomes will be measured at five time points for the treatment group (weeks 0, 4, 8, 12, and 28) and twice for the TAU control group (weeks 0 and 12). See Table 1 for the participants’ assessment schedule. Of note, there will be a differential assessment schedule for TAU versus HRTP participants. That is, HRTP participants are assessed on three more occasions than those in the TAU group. Two of these additional assessment periods are included within the intervention arm’s active treatment phase (weeks 4 and 8) to better understand how each HRTP phase impacts healing/racial trauma symptom severity. Week 4 roughly translates to the participants transitioning to the Healing phase and week 8 the Empowerment phase. Further, those in the control arm have no need for a post-treatment follow-up as upon receiving TAU, they may be immediately enrolled within the treatment group to receive the HRTP.

The *Racial Trauma Scale* (RTS; [55]) is a 30-item clinical measure which examines the severity of traumatic symptoms due to racism. More specifically, the RTS asks individuals to “think about all the times when you have heard about, seen, or experienced racial discrimination” and how much they have been bothered by resultant thoughts, behaviours, or symptoms on a 4-point scale (where 1 is “not at all” and 4 is “extremely”). Of note, within this study specifically, when completing the RTS, clients will be asked to rate symptoms as per their lived experience over the previous month/30 days. An RTS total score is calculated by summing all item ratings together. Moreover, subfactors of the RTS include “Lack of Safety”, “Negative Cognitions”, and “Difficulty Coping”. Higher RTS total and subfactor scores indicate greater symptom distress. In consideration of the RTS’s convergent validity, RTS responses were found to be highly correlated with several measures of trauma, including the *PTSD Checklist for the DSM-5* (PCL-5; r = 0.79) and the *General Ethnic Discrimination Scale* (r = 0.76). The RTS was developed for and tested with diverse ethnoracial populations. The RTS also has excellent reliability (α = 0.97; [55]).

The *Beck Depression Inventory* (BDI-II; [3]) is a 21-item clinical measure used to evaluate the severity of depressive symptoms in populations aged 13+. In so doing, respondents are asked to consider 21 groups of statements and choose the statement that most aligns with their experience over the previous 2-week period. Each item’s statements exist on a scale running from 0 to 3. For example, on the item of “self-criticalness” respondents select from the following group of related statements “0—I don’t criticize or blame myself more than usual; 1—I am more critical of myself than I used to be; 2—I criticize myself for all of my faults; 3—I blame myself for everything bad that happens”. A total score is then calculated by summing up all items’ corresponding statement ratings, with higher scores suggestive of a greater severity of depressive symptoms. Total BDI–II scores within the 0–13 range are indicative of no or minimal depression, 14–19 mild depression, 20–28 moderate depression, and 29–63 severe depression. The test–retest reliability of the BDI-II is between 0.73 and 0.96 ([47]). Moreover, convergent validity was tested against several validated measures of depression, with correlation coefficients ranging from 0.57 to 0.94. Also, [37] ([37]) found no evidence of racial bias in the BDI-II.

Finally, the *Trauma Symptoms of Discrimination Scale* (TSDS; [59]) is a two-part clinical measure of discriminatory distress. In the first part, which is 21 items, individuals characterize the severity of anxiety-related trauma symptoms including avoidance, negative cognitions, social fears, and worries about the future. Items are rated on a 4-point scale (where 1 is “never” and 4 is “often”), and higher TSDS scores indicate greater distress caused by discriminatory acts. Of note, within this study specifically, when completing the TSDS, clients will be asked to rate distress as per their lived experience over the previous month/30 days. In the TSDS’ second part, individuals are asked to select what form of discrimination they have experienced, including racial/ethnic, gender, sexual orientation, religion, age, and/or other types of discrimination. Moreover, individuals can check multiple forms of discrimination. The TSDS was tested against several known measures of psychopathology and experiences of racism, including PTSD symptoms (PCL-5; *r* = 0.69), traumatic cognitions (*Post Traumatic Cognitions Inventory*; *r* = 0.65) and racial trauma (*Race-Based Traumatic Stress Symptom Scale—Short Form*; *r* = 0.69). The TSDS was developed for and tested with diverse ethnoracial groups ([57]).

### 2.10. Secondary Outcomes

Two secondary outcomes will also be measured at three time points within the treatment group only (weeks 4, 8, and 12). See Table 1 for the participants’ assessment schedule.

The *Helping Alliance Questionnaire* (HAq-II; [26]) is a 19-item clinical measure which asks respondents to consider their relationship with their therapist, and rate items according to how strongly they agree or disagree. Sample items include “I feel I am working together with my therapist in a joint effort” and “The therapist and I make unprofitable exchanges”. Items are rated on a 6-point scale, where “1” is strongly disagree and “6” is strongly agree. A total score is calculated from a sum of all items (with some items requiring reverse scoring first). Higher scores are indicative of a better-quality therapeutic alliance between the respondent and the therapist, as perceived by the respondent. The HAq-II has a test–retest reliability over a three-session span of 0.78. It has previously been tested with African American adult clients in a substance use treatment setting, finding comparable conceptualizations of the therapeutic alliance ([16]).

The *Racial Microaggressions in Counselling Scale* (RMCS; [15]) is a 10-item clinical measure designed to assess the occurrence of subtle covert expressions of prejudice or racism (i.e., microaggressions) within the therapeutic setting as perceived by the respondent/client. Sample items include “my counsellor avoided discussing or addressing cultural issues in our session(s)” and “my counsellor sometimes seemed unaware of the realities of race and racism”. Items are rated on a 3-point scale where “0” is this never happened and “2” this happened and I was bothered by it. An RMCS total is obtained by summing all item ratings together, with higher total scores suggestive of more microaggressions being observed within the therapeutic setting. The RMCS was developed for and tested within diverse ethnoracial groups.

### 2.11. Feasibility for Wider Application

Prior to the RCT proposed herein, a pilot study of a similar intervention arm design was undertaken. That is, early career clinicians and clinical trainees administered the HRTP to racialized clients under the supervision of clinical psychologists. Below is feasibility data from the pilot study’s early clients and clinicians. Data collection for the pilot study was nearly complete at the time of this manuscript’s submission.

*Implementation* refers to the degree to which a new treatment can be successfully delivered within a given setting ([6]). Within the pilot study conducted preceding this proposed study, implementation was operationalized in terms of time to completion. That is, were therapists able to deliver the HRTP within a reasonable time frame? Within the pilot study, therapists took 13.66 weeks on average to complete the HRTP with clients (*n* = 10 completers), which is well within the expected 3–4-month time frame.

The team also considered *practicality* or the extent to which the new treatment can be carried out using existing means, absent external intervention. As such, practicality was operationalized in terms of (1) therapist ease in terms of adhering to the protocol without the need for deviation; and (2) client homework completion (as a proxy for the client’s abilities to carry out important protocol activities). The said elements were largely measured using session notes. More specifically, therapists completed session progress notes within 24 h of each HRTP session. Each note asked therapists to reflect on several items, including but not limited to, “Were you required to deviate from the treatment protocol approved agenda in any way during the session? If so, please explain how you deviated and why?” There is also space for the therapist to indicate homework completion for the session on a 4-point scale: “completed”; “partially completed”; “not attempted”; or “not applicable”.

On therapist adherence to the assigned protocol, within the pilot study (*n* = 14 client cases which included 10 completers and 4 early dropouts) 71.4% of sessions in the Stabilization phase, 66.7% in the Healing phase, and 62.5% in the Empowerment phase required no deviation on the part of therapists to meet client needs. Moreover, it is important to note, many of the deviations reported related to either (i) allocating more or less time to an existing protocol agenda item than was suggested in the protocol; or (ii) rearranging protocol agenda items to better fit the natural flow of the session. Few of the deviations reported included the therapist feeling obligated to look beyond the protocol or introduce non-HRTP interventions to meet client needs (between 9.5 and 17.6%, depending on the treatment phase). On the matter of client homework completion, clients in the pilot study had a completion rate of 51.2% in the Stabilization phase, 56.9% in the Healing phase, and 62.5% in the Empowerment phase. There was also an additional 12.2%, 35.3%, and 32.5% partial completion rate for the different phases, respectively.

Conversely, *acceptability* is defined as the perception of the new treatment by implementation stakeholders ([6]). As such, acceptability was operationalized in terms of (1) therapist comfort with delivering the protocol, as well as (2) client satisfaction with the HRTP. On therapist comfort, part of each session note reflection required therapists to rate their level of comfort in terms of delivering the session material on a 5-point scale: “1—extremely uncomfortable”, “2—slightly uncomfortable”, “3—neither comfortable or uncomfortable”, “4—slightly comfortable”, or “5—extremely comfortable”. And finally, as an element of each client’s week 20 data collection (note, within the pilot study we conducted follow-up at 20 weeks, not 28), clients were asked “If you had a family member or close friend that suffered from racial trauma, would you recommend they complete the HRTP. Why or why not?”.

On therapist comfort with delivering the HRTP (*n* = 14 client cases which included completers and early dropouts), the mean comfort rating was 4.25 (*SD* = 1.01) within the Stabilization phase, 4.59 (*SD* = 0.83) in the Healing phase, and 4.60 (*SD* = 0.81) in the Empowerment phase. As such, throughout the delivery of the HRTP, the therapists’ comfort levels averaged somewhere between “slightly comfortable” and “extremely comfortable”. On whether or not pilot study clients opted to recommend the HRTP to family and friends, 100% of the completers who participated in an exit questionnaire (*n* = 7 completers) noted that they “would recommend” the protocol to loved ones in future.

### 2.12. Statistical Analysis

Descriptive statistics, including means, standard deviations, and frequencies, will be calculated for continuous primary and secondary outcomes. Differences in primary and secondary outcomes, including TSDS, RTS, change in symptoms, and therapeutic experiences, will be examined using a two-way mixed ANOVA with two or three within-subject levels of time (pre-test, post-test, and follow-up) and two between-subject levels of condition (HRTP and control). When there is evidence of sphericity violation, we will use the Greenhouse–Geisser correction ([22]). It is predicted that participants’ TSDS and RTS scores will decrease significantly over the course of the treatment. It is also predicted that TSDS and RTS scores at follow-up will be the same or lower than at the end of the treatment.

Both the short-term and long-term efficacy of the HRTP will be examined during the treatment (week 0, 4, 8), at the end of the treatment (week 12), and 4 months after the treatment (week 28). Assessment will take place at four time points for the treatment group (HRTP therapy; weeks 0, 4, 8, 12, and 28; see Table 1 for participants’ assessment schedule) and only two time points for the control TAU group (T0 and T4; see Table 1 for participants’ assessment schedule). Analysis will compare standard mean differences (SMDs) between the time points for the treatment group (weeks 0, 12, and 28) and the control group (weeks 0 and 12). Analyses will entail intent to treat (ITT) and include participants with pre-treatment and the last time point available as long as they attended at least one treatment session. We examine significant levels for the analysis as well as effect sizes. A significance level of *p* < 0.05 will be used for all tests.

For each outcome of interest, we will identify any significant Condition × Time interactions. When no interaction is found, significant main effects will be reported. Partial eta-squared (η2) was used to measure effect sizes for the ANOVAs. When significant interactions are found, post hoc tests of simple main effects will be used to examine the differences, *p* values will be included for all post hoc tests, and 95% confidence intervals (CIs) will be used for all *p* values less than 0.05. Additionally, we will conduct comparisons of baseline data for completers and non-completers. Subgroup analysis will be based on age, gender, and levels of racial stress and trauma.

The secondary outcomes will compare the quality of the therapeutic experiences throughout the treatment group (HRTP therapy; weeks 4, 8, 12), as well as comparing the quality of the sessions between the treatment and control groups (week 12). The strength of the therapeutic alliance between the participants and therapists (Helping Alliance Questionnaire; [26]) will be evaluated, as will whether or not the participants perceived any microaggressions in their sessions (Racial Microaggressions in Counselling Scale; [15]).

## 3. Discussion

Racialized Canadians face mental health disparities due the historical and ongoing effects of racism and colonization, which is further exacerbated by barriers in accessing care. When they access care, many mental health providers are not adequately prepared to address the intersection of racial trauma and systemic oppression, which affect racialized individuals ([2]). This paper presents the design and methodology of a randomized controlled trial on the efficacy of the newly developed *Healing Racial Trauma Protocol* (HRTP) for people of colour suffering from racial trauma compared to a control condition of treatment-as-usual (TAU) in reducing the severity of racial trauma and related symptoms, as well as improving functioning. The preliminary feasibility evidence presented shows the larger implementation of this protocol as feasible, scalable, and acceptable for clients and therapists. This paper also explored several methodological considerations related to trial design and research population. Overall, the proposed design can serve as inclusive and viable models for engaging in clinical research related to racialized populations.

This paper is timely given the ongoing and disproportionate exclusion of many racialized groups—including Black and Indigenous populations—from research and clinical trials (e.g., [32]; [45]). Despite the disproportionately high rates of trauma in these communities ([21]), current methodological and research practices limit the generalizability of existing research and treatment evidence for PTSD and other mental health conditions to racialized communities ([21]; [28]; [34]). This paper addresses these inequities by presenting strategies for inclusive, equitable, and culturally informed approaches to researching and treating racial trauma. A primary strategy is the diversification of the research team to include racialized experts in key leadership and outreach roles ([48]), as well as representation across a range of identities and experiences (e.g., sexual identity, age, religious beliefs, clinical training) to ensure the team can work effectively with a range of research participants. To support outreach, knowledge sharing and dissemination are critical to ensure racialized communities have access to information about the trial and the ways in which they can participate. Given the concerns about research safety and potential abuses, adequate recruitment requires open, clear, and direct communication channels with diverse communities providing ample details about the study and psychoeducation to promote trust and transparency. In addition to sufficient information about the study, making meaningful personal connections with trusted community partners, leaders, and members are essential ([48]). The recruitment and outreach activities for the HRTP trial align with these best practices.

### Implications

A protocol such as this is urgently needed as we better understand the impact of racism on the health of people of colour and what can be done about it. Racial trauma can have long-term negative effects on individuals, but at the moment it is underrecognized and undertreated. The majority of mental health providers in Canada are White ([18]; [36]), and as such may have a limited understanding of the experiences and impacts of racism. This lack of understanding can lead to a lack of cultural competence and an inability to effectively address and treat racial trauma in therapy.

The most immediate implications of this study will be used to validate, refine, and ultimately strengthen the HRTP for use within racialized communities. Initial evidence from this trial can support the validity and efficacy of the HRTP in reducing symptoms of racial trauma in community and clinical settings. Additionally, the experiences and feedback of the participants, therapists, and research team can be used to refine and improve the protocol’s content and administration. Overall, study findings could be used to inform decisions on the implementation of this protocol in other community and clinical settings.

This trial is essential for advancing clinical knowledge and practice to effectively support the many individuals and communities of colour affected by racial trauma, including traumatic experiences suffered in society and in the workplace (e.g., [5]). In fact, given the cultural and racial diversity of Canada, we are uniquely positioned to lead the development of the first national and international guidelines for the treatment of racial trauma. Thus, the HRTP has the potential to be a much-needed tool for mental health providers treating racially traumatized clients across Canada and abroad.

## Figures and Tables

**Figure 1 behavsci-15-00856-f001:**
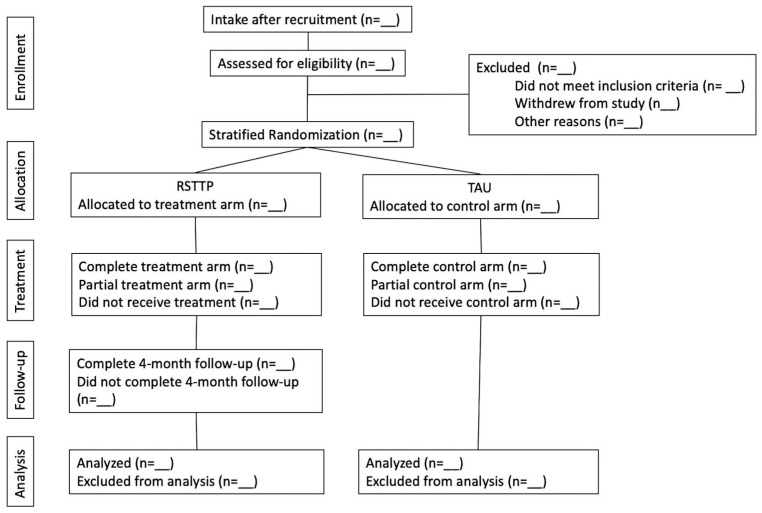
Participant flow diagram in HRTP Trial.

**Table 1 behavsci-15-00856-t001:** Participant assessment schedule.

Time	Session	Week	Measures	
			Intervention Group	Control/TAU Group
T1	Intake	0	MINI; UnRESTS; BDI-II; RTS; TSDS	
T2	4	4	BDI-II; RTS; TSDS; HAq-II; RMCS	nothing
T3	8	8	BDI-II; RTS; TSDS; HAq-II; RMCS	nothing
T4	12	12	BDI-II; RTS; TSDS; HAq-II; RMCS	BDI-II; RTS; TSDS; HAq-II; RMCS
T5	Follow Up	28	BDI-II; RTS; TSDS	participants will have withdrawn from the study or started the protocol

Note. BDI-II: Beck Depression Inventory-II; HAq-II: Helping Alliance Questionnaire-II; MINI: The Mini International Neuropsychiatric Interview; RMCS: Racial Microaggressions in Counselling Scale; RTS: Racial Trauma Scale; TSDS: Trauma Symptoms of Discrimination Scale; UnRESTS: University of Connecticut Racial/Ethnic Stress & Trauma Scale.

## Data Availability

The research data, analytical methods, and study materials can be available upon request to interested researchers by contacting Dr. Monnica Williams at monnica.williams@uottawa.ca.

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
