# Peer review of "Treating Racial Trauma: The Methodology of a Randomized Controlled Trial of the Healing Racial Trauma Protocol"

_behavsci, 2025, doi:10.3390/bs15070856_

Round 1

Reviewer 1 Report

Comments and Suggestions for Authors

Reviewer Comments

Thank you for the opportunity to review the manuscript titled “Treating racial trauma: Methodology of a randomized con- 2 trolled trial of the Healing Racial Trauma Protocol.” The manuscript provides informed evidence regarding rationale for an experimental study to evaluate an intervention regarding racial trauma, that is, Healing Racial Trauma Protocol (HRTP). The manuscript also provides reasonable protocol for the experimental study. However, to further improve the utility and impact of the manuscript, along with strategies to further improve the proposed trial, I suggest considering following minor changes/additions. I also have a few editorial suggestions; however, feel free to disregard those if they do not align with the Canadian English structure and grammar

  1. Introduction: This section systematically lays out the rationale for the intervention and the experimental trial. However, below are a few suggestions:
    1. Line 129: Please include reference/s for “research studies” (line 127) via which HRTP has been vetted.
    2. Lines 129-132: I appreciate the authors highlighting the ideas of ongoing education, cultural competency, and anti-bias work in fostering therapeutic relationships. However, it will be beneficial to readers if authors can also highlight in this section the limitations associated if these approaches to address the desired outcome of therapeutic relationships (for e.g., Misra et al., 2021; Onyeador et al., 2021; Truong et al., 2014). While these interventions are frequently employed, existing evidence highlight limitations in these approaches.
    3. Lines 138: Consider using “more effective” instead of “deeper” since deeper can be interpreted in a variety of ways.
  2. Methods: This section lays out the methodological information fairly well. However, below are a few recommendations for the authors to consider.
    1. Please include information regarding incentive for participation. I understand that specific incentive information may not be available yet; however, please highlight if there will be any incentives for the participants (participants and the therapists) in this study.
    2. I appreciate that the authors have included extensive information regarding factors associated with participation among racialized individuals. However, please include some information regarding how the idea of international migration will be assessed or addressed in this study. This is particularly important since migration can either protect from or introduce risk of racial trauma. This issue can be further complicated based on the generational cohort of the immigrants (1st 2nd generation migrants). While the random assignment and stratified sampling may alleviate some of these confounding variables, a more informed approach might have more utility and safeguard the study from Type 1 or Type 2 error.
    3. Line 229: Readers will appreciate any examples of how the research team may address “barriers to participation” and challenges related to attendance. For e.g., will the research team provide transportation to participants if they prefer to meet in-person but have no means for transportation or limited knowledge regarding/preference for telehealth modalities? Examples will also help better assess feasibility of the trial.
    4. Line 234: Please change “rial” in the sentence to “trial.”
    5. Figure 1: Under the block of “Follow-up”, the figure highlights “Complete 3-month follow-up”, which is confusing since the authors consistently suggest 4-month follow-up in the manuscript text.
    6. Line 244 Exclusion criteria: I appreciate the authors for outlining the criteria for study participation. To further reduce confounding variables, I wonder if authors have considered exclusion based on physical disability. Given that physical disability has an intersecting layer of stigma and discrimination with mental health concerns and racial trauma, it might be prudent to consider exclusion based on physical health concerns.
    7. Line 268: I am curious about how the authors aim to address self-selection bias, since the time commitment needed for this study is extensive. From the start, not everybody has access to a 3-hour window for an intake session so people who are economically safe and do not have multiple employments are more likely to participate in this intervention, which can be a limitation here, since individuals who are economically struggling tend to have more risk for and exposure to racial trauma.
    8. Line 272: It is not clear from the text if the authors will create “files” using real names or pseudonyms. It might be good to clarify this for the readers, in case readers have any concerns regarding participant protection/privacy and data security. In line with this query, please also include information regarding how the authors aim to securely store data. Given that the HRTP intervention is primarily for racialized individuals and will include their private conversations regarding racial trauma, it is crucial to outline how the participants’ data will be securely collected, stored, and analyzed.
    9. Line 297: Please consider changing “participants” at the end of the sentence to “participation” because currently the sentence ends as “participants to sustain their participants.”
    10. Line 329: Please consider adding “the” before “said clinic.”
    11. Line 357-359: It is not clear why all the therapists in the intervention arm will not be required to read and reflect on the readings. To address confounding variables related to therapists’ participation in this study, I suggest either requiring every therapist to read and reflect on the articles so that there is consistency or consider establishing benchmarks based on which “minimal clinical experience overall or in working with racialized clients” will be ascertained. An entirely subjective decision making regarding the idea of “minimal” can lead to an inadvertent confounding variable within this study.
    12. Line 364-370: In line with the previous comment, I suggest establishing a short quantitative survey with a cut-off score to determine engagement with the material instead of a short written or oral reflection that is evaluated by only one personnel (Project Coordinator). If the authors believe that a quantitative survey will be ineffective or redundant here then I suggest including another personnel to review therapists’ engagement with the material to limit influence of biases from review by only one personnel.
    13. Line 606: Please consider revising “Analyses were intent-to-treat (ITT) and included participants” to “Analyses will entail intent-to-treat (ITT) and include participants…” since the trial will occur in future.
  3. Discussion section: My final comment is regarding the general idea of an intervention that is aimed at addressing racial trauma without explicitly addressing systemic or policy level factors. This can be regarded by some as a significant limitation for an intervention such as HRTP. For e.g., how can this intervention address or prevent racial trauma if an individual experiences racism 6 months after receiving HRTP as an intervention? While I agree with the rationale for the intervention, the authors might want to include some information in the discussion section on how an intervention that is aimed to address concerns at an individual or client-provider level can help in addressing or minimizing effects of the systemic factors. Some discussion regarding this aspect will bolster both the rationale and the utility of this intervention.

References

Misra, S., Jackson, V. W., Chong, J., Choe, K., Tay, C., Wong, J., & Yang, L. H. (2021). Systematic review of cultural aspects of stigma and mental illness among racial and ethnic minority groups in the United States: Implications for interventions. American Journal of Community Psychology68(3-4), 486-512.

Onyeador, I. N., Hudson, S. K. T., & Lewis Jr, N. A. (2021). Moving beyond implicit bias training: Policy insights for increasing organizational diversity. Policy Insights from the Behavioral and Brain Sciences8(1), 19-26.

Truong, M., Paradies, Y., & Priest, N. (2014). Interventions to improve cultural competency in healthcare: A systematic review of reviews. BMC health services research14, 1-17.

Comments on the Quality of English Language

I have provided a few editorial suggestions; however, feel free to disregard them if they do not align with the Canadian English structure or grammar.

Author Response

Revision of Treating racial trauma: Methodology of a randomized controlled trial of the Healing Racial Trauma Protocol

Responses to Reviewer 1

Please see our replies to reviewer 1 below and the edits in the manuscript are highlighted in yellow.

Reviewer 1 Comments

Thank you for the opportunity to review the manuscript titled “Treating racial trauma: Methodology of a randomized controlled trial of the Healing Racial Trauma Protocol.” The manuscript provides informed evidence regarding rationale for an experimental study to evaluate an intervention regarding racial trauma, that is, Healing Racial Trauma Protocol (HRTP). The manuscript also provides reasonable protocol for the experimental study. However, to further improve the utility and impact of the manuscript, along with strategies to further improve the proposed trial, I suggest considering following minor changes/additions. I also have a few editorial suggestions; however, feel free to disregard those if they do not align with the Canadian English structure and grammar

Comment 1: Introduction: This section systematically lays out the rationale for the intervention and the experimental trial. However, below are a few suggestions:

Response 1: We thank Reviewer 1 for their detailed comments on the introduction. The suggested revisions were made to the revised manuscript and highlighted in yellow.

1.1 Line 129: Please include reference/s for “research studies” (line 127) via which HRTP has been vetted.

Response 1.1: Thank you for this suggestion. We have added two relevant citations (Lines 123-127).

1.2 Lines 129-132: I appreciate the authors highlighting the ideas of ongoing education, cultural competency, and anti-bias work in fostering therapeutic relationships. However, it will be beneficial to readers if authors can also highlight in this section the limitations associated if these approaches to address the desired outcome of therapeutic relationships (for e.g., Misra et al., 2021; Onyeador et al.,2021; Truong et al., 2014). While these interventions are frequently employed, existing evidence highlight limitations in these approaches.

Response 1.2: Thank you for sharing these additional citations. Although we found these citations informative, given the population, focus, and context of these citations, we have decided not to include them in the manuscript. Misra and colleagues’ article is not specifically relevant to the limitations associated with cultural competency and anti-bias work on the part of therapists (Misra et al., 2021). In fact, this paper mentioned the value of training providers and integrating cultural considerations in health care provision for managing stigma (Misra et al., 2021). Truong and colleagues’ concept of cultural competency, including culturally tailored health promotion workshops, is very different from the concept of cultural safety and competency in our study, such focuses on building therapeutic relationships during clinical sessions over several weeks. The authors found culturally tailored health education program did not shift client health behaviors suggesting culturally tailored health promotion workshops have limited impact (Truong et al., 2014). Finally, the Onyeador and colleagues, highlight how anti-bias training can be inefficient in shifting organizational structures and promoting behavioral change in large organizations (Onyeador et al., 2021). In fact, the authors recommend a range of structural changes to promote equity within organizations, beyond simply training staff.

1.3 Lines 138: Consider using “more effective” instead of “deeper” since deeper can be interpreted in a variety of ways.

Response 1.3: Thank you for this revision. We have made the change to improve clarity (Line 138).

Comment 2: Methods: This section lays out the methodological information fairly well. However, below are a few recommendations for the authors to consider.

2.1 Please include information regarding incentive for participation. I understand that specific incentive information may not be available yet; however, please highlight if there will be any incentives for the participants (participants and the therapists) in this study.

Response 2.1: Thank you for this comment. Indeed, participants, therapists, and supervisors in this study would be compensated for their efforts. The revised version of the manuscript now says so within the “Therapist Eligibility, Training and Supervision” and “Participant Retention Strategies” sections (Lines 315-318; 344-346; 353 - 355).

2.2 I appreciate that the authors have included extensive information regarding factors associated with participation among racialized individuals. However, please include some information regarding how the idea of international migration will be assessed or addressed in this study. This is particularly important since migration can either protect from or introduce risk of racial trauma. This issue can be further complicated based on the generational cohort of the immigrants (1st 2nd generation migrants).

While the random assignment and stratified sampling may alleviate some of these confounding variables, a more informed approach might have more utility and safeguard the study from Type 1 or Type 2 error.

Response 2.2: Thank you for this comment. We recognize the nature, scope, and intensity of traumatic experiences are varied and might have differential impact on racial trauma. However, we will test the extent to which the strategies and skills outlined in the HRTP are beneficial and result in reductions of racial trauma for a stratified sample focusing on age and gender differences. Given the limited research in this area, the extent to which different demographic characteristics and life conditions could impact experiences of racial trauma remain to be examined.

2.3 Line 229: Readers will appreciate any examples of how the research team may address “barriers to participation” and challenges related to attendance. For e.g., will the research team provide transportation to participants if they prefer to meet in-person but have no means for transportation or limited knowledge regarding/preference for telehealth modalities? Examples will also help better assess feasibility of the trial.

Response 2.3: Thank you for this suggestion. We have added to the section on “Participant Retention Strategies” (Lines 307 - 313).

2.4 Line 234: Please change “rial” in the sentence to “trial.”

Response 2.4: Thank you for highlighting this typo. We have made this change.

2.5 Figure 1: Under the block of “Follow-up”, the figure highlights “Complete 3-month follow-up”, which is confusing since the authors consistently suggest 4-month follow-up in the manuscript text.

Response 2.5: Thank you for highlighting this discrepancy. The last time point in the design is 4 months (16 weeks) after the completion of the treatment. We have updated the Figure 1.

2.6 Line 244 Exclusion criteria: I appreciate the authors for outlining the criteria for study participation. To further reduce confounding variables, I wonder if authors have considered exclusion based on physical disability. Given that physical disability has an intersecting layer of stigma and discrimination with mental health concerns and racial trauma, it might be prudent to consider exclusion based on physical health concerns.

Response 2.6: Thank you for highlighting this additional demographic and identity consideration. The location of the study is accessible to individuals with physical disabilities. It is not feasible at this point to sufficiently account for a range of demographics characteristics that might mental health experiences. We will use a stratified randomization process to incorporate study several covariates (i.e., age, gender, racial stress levels).

2.7 Line 268: I am curious about how the authors aim to address self-selection bias, since the time commitment needed for this study is extensive. From the start, not everybody has access to a 3-hour window for an intake session so people who are economically safe and do not have multiple employments are more likely to participate in this intervention, which can be a limitation here, since individuals who are economically struggling tend to have more risk for and exposure to racial trauma.

Response 2.7: Thank you for this comment. We acknowledge selection bias as a limitation, but this would be the case for most any psychotherapy treatment studies (e.g., Foa & Williams, 2010). We have included this in the text (Lines 265-267). Also, we are paying participants for their time to offset the risk that this will be an issue.

2.8 Line 272: It is not clear from the text if the authors will create “files” using real names or pseudonyms. It might be good to clarify this for the readers, in case readers have any concerns regarding participant protection/privacy and data security. In line with this query, please also include information regarding how the authors aim to securely store data. Given that the HRTP intervention is primarily for racialized individuals and will include their private conversations regarding racial trauma, it is crucial to outline how the participants’ data will be securely collected, stored, and analyzed.

Response 2.8: The participant charts use their real names and are stored in an ERM system with the same security as medical patient files. And data compiled into a dataset for analysis would only include participant numbers and not names. No names will be included in publications, as data will be reported in aggregate. These details are included in the text (Line 285 - 289).

2.9 Line 297: Please consider changing “participants” at the end of the sentence to “participation” because currently the sentence ends as “participants to sustain their participants.”

Response 2.9: Thank you for your keen eye. We have made the recommended change.

3 Line 329: Please consider adding “the” before “said clinic.”

Response 2.10: Changed as requested.

3.1 Line 357-359: It is not clear why all the therapists in the intervention arm will not be required to read and reflect on the readings. To address confounding variables related to therapists’ participation in this study, I suggest either requiring every therapist to read and reflect on the articles so that there is consistency or consider establishing benchmarks based on which “minimal clinical experience overall or in working with racialized clients” will be ascertained. An entirely subjective decision making regarding the idea of “minimal” can lead to an inadvertent confounding variable within this study.

Response 2.11: The treatment-as-usual group should reflect what therapists would typically do. If we provide articles about anti-racist clinical practice to the TAU group, that will be a study confound.

3.2 Line 364-370: In line with the previous comment, I suggest establishing a short quantitative survey with a cut-off score to determine engagement with the material instead of a short written or oral reflection that is evaluated by only one personnel (Project Coordinator). If the authors believe that a quantitative survey will be ineffective or redundant here then I suggest including another personnel to review therapists’ engagement with the material to limit influence of biases from review by only one personnel.

Response 2.12: Thank you for this feedback, we have reviewed the section and updated the process to include the Project Coordinator and at least one other HRTP supervisor (Lines 373 - 395).

3.3 Line 606: Please consider revising “Analyses were intent-to-treat (ITT) and included participants” to “Analyses will entail intent-to-treat (ITT) and include participants…” since the trial will occur in future.

Response 2.13: Thank you for highlighting this typo. We have made this change.

Comment 3: Discussion section: My final comment is regarding the general idea of an intervention that is aimed at addressing racial trauma without explicitly addressing systemic or policy level factors. This can be regarded by some as a significant limitation for an intervention such as HRTP. For e.g., how can this intervention address or prevent racial trauma if an individual experiences racism 6 months after receiving HRTP as an intervention? While I agree with the rationale for the intervention, the authors might want to include some information in the discussion section on how an intervention that is aimed to address concerns at an individual or client-provider level can help in addressing or minimizing effects of the systemic factors. Some discussion regarding this aspect will bolster both the rationale and the utility of this intervention.

Response 3: The HRTP protocol includes psychoeducation about the nature of systemic racism in Part 1, practice for addressing racism in real life in Part 2, and support for reducing systemic oppression in Part 3. The clinician cannot reduce the racism in the lives of participants, but can help them address the components of their lives that can be changed and manage those things that cannot.

References

Misra, S., Jackson, V. W., Chong, J., Choe, K., Tay, C., Wong, J., & Yang, L. H. (2021). Systematic review of cultural aspects of stigma and mental illness among racial and ethnic minority groups in the United States: Implications for interventions. American Journal of Community Psychology, 68(3-4), 486-512.

Onyeador, I. N., Hudson, S. K. T., & Lewis Jr, N. A. (2021). Moving beyond implicit bias training: Policy insights for increasing organizational diversity. Policy Insights from the Behavioral and Brain Sciences, 8(1), 19-26.

Truong, M., Paradies, Y., & Priest, N. (2014). Interventions to improve cultural competency in healthcare: A systematic review of reviews. BMC health services research, 14, 1-17

Reviewer 2 Report

Comments and Suggestions for Authors

This is a well written and well supported fascinating paper on an important topic. Overall, this is a convincingly presented, clearly well researched, detailed discussion of the methodology of a RCT. I only have a few comments for consideration.

One suggestion for consideration is the inherent contradictions in some of the research literature presented - e.g. on p2 lines 59 onwards seems to suggest that mental health issues as a result of racism are under reported and diagnosed but generally that racialised minorities suffer from an over diagnosis of mental health issues, and then again that there is a lack of reliable data - is this a contradiction should the gaps in the literature and the lack of clarity be highlighted here?  

Sample size seems low to me. 

On p6 line 252 - check sentences starting 'Said...'

Could the additional support provided to HRTP therapists lead to overestimating the treatment effect? This additional support - selection process, training, weekly support would presumably not be available if this therapy is rolled out more generally?

Explain the different assessment schedules for TAU and HRTP groups and their possible effects.

Author Response

Revision of Treating racial trauma: Methodology of a randomized controlled trial of the Healing Racial Trauma Protocol

Responses to Reviewer 2

Please see our replies to reviewer 2 below and the edits in the manuscript are highlighted in yellow.

Reviewer 2 Comments

This is a well written and well supported fascinating paper on an important topic. Overall, this is a convincingly presented, clearly well researched, detailed discussion of the methodology of a RCT. I only have a few comments for consideration.

Comment 1: One suggestion for consideration is the inherent contradictions in some of the research literature presented - e.g. on p2 lines 59 onwards seems to suggest that mental health issues as a result of racism are under reported and diagnosed but generally that racialised minorities suffer from an over diagnosis of mental health issues, and then again that there is a lack of reliable data - is this a contradiction should the gaps in the literature and the lack of clarity be highlighted here?  

Response 1: Most racialized groups are overdiagnosed with certain disorders (psychosis, antisocial personality) and underdiagnosed with others (anxiety, depression). These under/overdiagnoses are due to false stereotypes and inadequate clinician training. We have clarified this in the text (Line 60-63).

Comment 2: Sample size seems low to me.

Response 2: The sample estimation was based on the previous studies on racial trauma and the expected standard mean difference in the outcome between the two arms. There are psychotherapy trials with similar sample sizes. Given there are no validated treatment protocols for racial trauma, this initial study will test the preliminary efficacy of this protocol.  

Comment 3: On p6 line 252 - check sentences starting 'Said...'

Response 3: Fixed. Thank you.

Comment 4: Could the additional support provided to HRTP therapists lead to overestimating the treatment effect? This additional support - selection process, training, weekly support would presumably not be available if this therapy is rolled out more generally?

Response 4: Typical treatment outcome studies provide training to therapists and ongoing supervision to ensure fidelity to the treatment model (e.g., Sloshower et al., 2023).

Sloshower, J., Guss, J., Krause, R., Wallace, R., Williams, M., Reed, S., & Skinta, M. (2020). Psilocybin-assisted therapy of major depressive disorder using Acceptance and Commitment Therapy as a therapeutic frame. Journal of Contextual Behavioral Science, 15, 12-19.

Comment 5: Explain the different assessment schedules for TAU and HRTP groups and their possible effects.

 Response 5: The different assessment schedules are now explained within “Primary Outcomes” (Lines 481- 488).
